# A Multi-Tissue Gene Expression Atlas of Water Buffalo (*Bubalus bubalis*) Reveals Transcriptome Conservation between Buffalo and Cattle

**DOI:** 10.3390/genes14040890

**Published:** 2023-04-10

**Authors:** Jingfang Si, Dongmei Dai, Kun Li, Lingzhao Fang, Yi Zhang

**Affiliations:** 1College of Animal Science and Technology, China Agricultural University, Beijing 100193, China; sijingfang@cau.edu.cn (J.S.); daidm@cau.edu.cn (D.D.); lk13488330733@163.com (K.L.); 2The Center for Quantitative Genetics and Genomics (QGG), Aarhus University, 11, 8000 Aarhus, Denmark

**Keywords:** RNA-seq, gene expression atlas, tissue-specific gene, house-keeping gene, comparative transcriptome

## Abstract

We generated 73 transcriptomic data of water buffalo, which were integrated with publicly available data in this species, yielding a large dataset of 355 samples representing 20 major tissue categories. We established a multi-tissue gene expression atlas of water buffalo. Furthermore, by comparing them with 4866 cattle transcriptomic data from the cattle genotype–tissue expression atlas (CattleGTEx), we found that the transcriptomes of the two species exhibited conservation in their overall gene expression patterns, tissue-specific gene expression and house-keeping gene expression. We further identified conserved and divergent expression genes between the two species, with the largest number of differentially expressed genes found in the skin, which may be related to structural and functional differences in the skin of the two species. This work provides a source of functional annotation of the buffalo genome and lays the foundations for future genetic and evolutionary studies in water buffalo.

## 1. Introduction

Gene expression atlases have been widely used to investigate gene expression in different tissues, cell types, and developmental stages. These resources provide a comprehensive view of gene expression patterns across the genome, which can help improve the functional annotation of the genome and understanding of the molecular mechanisms underlying different tissues and complex biological processes. In humans, the Functional Annotation of the Mammalian Genome Consortium (FANTOM) [1] and the Encyclopedia of DNA Elements project (ENCODE) [2] were proposed to facilitate the elucidation of numerous human disease genes and the identification of functional elements within the human genome. Numerous international consortium projects, such as the Genotype–Tissue Expression (GTEx) [3] and the International Human Epigenome Consortium (IHEC) [4], have been initiated with the objective of establishing the correlation between genetic variation and gene expression in human tissues and deciphering the epigenetic regulation of cell states that are relevant to human health and disease. Recently, with technological advancement and data accumulation, integration of large-scale multi-omics data has gradually been applied in the field of agricultural animals, including the CattleGTEx Project [5], the PigGTEx Project [6] and the construction of multi-tissue gene expression atlases in beef cattle [7] and pigs [8].

The Asian water buffalo (*Bubalus bubalis*) is a large-bodied member of the Bovini tribe that is an economically important provider of milk, meat, draught power, and leather in at least 67 countries on five continents [9,10]. Due to its natural adaptation to tropical and subtropical environments, the water buffalo has played a key role in the sustainable development of global agriculture [11]. Its global population of 204 million in 2021 showed an increase of 20.9% over the past two decades (http://www.fao.org/faostat/, accessed on 6 April 2023). Notably, the water buffalo produces milk with rich nutrients (e.g., high fat and protein contents) and unique flavors and is especially suitable for cheese production [12,13].

There are two types of domesticated water buffaloes which are interfertile [14] and taxonomically classified as separate species [14]—the swamp buffalo (*Bubalus carabanensis* or *Bubalus kerabau*) found in China and Southeast Asia and the river buffalo (*Bubalus bubalis*) with a broad geographical distribution from the Indian subcontinent to Italy, the Americas and Australia [10]. Previous transcriptome studies have reported a gene expression atlas for river buffalo [15] but not for swamp buffalo. Additionally, their data were aligned to a draft buffalo reference genome (the scaffold level) with a limited annotation of genes [16]. In this study, we newly generated 73 RNA-Seq data from 19 tissues in swamp buffalo and integrated them with 282 publicly available RNA-Seq data from 51 tissues in water buffalo. We conducted quality control, read alignment, gene expression quantification and further bioinformatic analyses using a unified pipeline and constructed a multi-tissue gene expression atlas for water buffalo. Furthermore, we compared the transcriptomes between water buffalo and cattle, revealing global conservation in gene expression between the two species.

## 2. Materials and Methods

### 2.1. RNA-Seq Samples 

We collected 73 samples from 19 tissues in four swamp buffalo. Total RNA was prepared using the TRIzol reagent in accordance with the manufacturer’s recommendation. RNA sequencing was performed using the Illumina NovaSeq 6000 platform (Illumina, San Diego, CA, USA) with paired-end reads of 150 bp length. These newly generated data were integrated with 282 publicly available buffalo RNA-Seq data downloaded from the European Nucleotide Archive (ENA).

We analyzed all the 355 RNA-seq data uniformly following the bioinformatic pipeline in the CattleGTEx [5]. First, we used Trimmomatic v0.39 [5] with parameters “adapters/TruSeq3-SE.fa:2:30:10 LEADING:3 TRAILING:3 SLIDINGWINDOW:4:15 MINLEN:36” to perform quality control of all reads. Then the clean reads were mapped to the buffalo reference genome (UOA_WB_1, GCF_003121395.1) using single or paired mapping modules of STAR v2.7.3a [17] with parameters “outFilterMismatchNmax 3, outFilterMultimapNmax 10 and outFilterScoreMinOverLread 0.66”. Finally, we kept samples with unique mapping rates ≥75% and obtained the normalized expression (TPM) of annotated genes using Stringtie (v2.1.1) [18].

Cattle RNA-seq samples were analyzed uniformly by the CattleGTEx consortium [5], and the normalized gene expression (TPM) data were obtained from https://cgtex.roslin.ed.ac.uk/downloads/, accessed on 12 January 2023. Ultimately, we obtained normalized gene expression values (TPM) for 355 and 4866 RNA-seq samples from 20 common tissues in buffalo and cattle, respectively. 

### 2.2. Genome Similarity, Transcriptome Similarity and Homologous Gene Identification between Buffalo and Cattle

The sequence similarity of the genome, as well as transcriptome (coding sequence region, CDS region) between the reference genomes of buffalo (UOA_WB_1) and cattle (ARS-UCD1.2), were analyzed by using MashMap [19] that implements an approximate algorithm. The homologous genes were identified by OrthoFinder, a software that identifies orthologous genes by integrating the bidirectional best-hit principle and analysis of phylogenetic trees of genes [20]. Genes were further classified into four categories: one-to-one homology, complex homology (including one-to-many, many-to-one, or many-to-many), no homology, and non-protein [21].

### 2.3. Sample Clustering

We used the function IntegrateData in the R package Seurat [22] to combine gene expression datasets of buffalo and cattle, following the approach presented in a comparative transcriptome study between humans and cattle [23]. This methodology not only corrects for technical variations but also aligns the shared gene expression features across the datasets [22]. The integrated dataset was used in the subsequent cluster analysis and detection of differentially expressed genes (DEGs) between species. Afterward, we performed the t-distributed stochastic neighbor embedding (t-SNE) using the R package Rtsne [24] with parameter “dim = 2, perplexity = 350, theta = 0.5” to map the samples to a two-dimensional space based on corrected expression values of orthologous genes. We calculated the median gene expression in each tissue in buffalo and cattle separately to represent the “true” expression of the particular tissue in each species. We then performed hierarchical clustering using the R package heatmap to explore the relationship of tissues in buffalo and cattle based on the median gene expression.

### 2.4. Detection of Tissue-Specific Genes (TSGs)

We utilized the R package Limma [25] to identify TSGs through differential expression analysis. Specifically, we employed the functions model. matrix, lmFit, contrasts. fit, eBayes, and topTable to assess the differences in gene expression between samples in a particular tissue and samples in the remaining tissues. To account for multiple testing, *p*-values were adjusted using the Benjamini and Hochberg method (FDR) [26]. We defined tissue-specific genes as those with log_2_(FC) > 1.5 and FDR < 0.05 [23].

### 2.5. Detection of House-Keeping Genes (HKGs)

We identified preliminary HKGs (pHKGs) by selecting genes that are expressed on average above a threshold of TPM > 1 across all tissues. In order to investigate the changes in the expression of each pHKG within the buffalo and cattle expression profiles, we have utilized the coefficient of variation (CV) to measure the degree of variation of gene expression for each gene [27]. We divided the pHKGs into low variable expression (CV ≤ first quartile), medium variable expression (first quartile < CV < third quartile), and highly variable expression (CV ≥ third quartile) based on the quartiles of the total distribution of CV values [7,8]. To investigate the functional differences between pHKGs with low, medium, and high expression variability, we performed GO enrichment analysis on the pHKGs using the R package clusterProfiler [28]. Moreover, the low variable expression pHKGs were further considered as HKGs and subdivided into three groups based on the average expression across all tissues, including low expression (1 < TPM ≤ 10), medium expression (10 < TPM ≤ 50), and high expression (TPM > 50) [7,8].

### 2.6. Detection of Differentially Expressed Genes (DEGs) between Species

To identify DEGs between species, we utilized the R package Limma [25] and then considered genes with log_2_(FC) > 1.2 and FDR < 0.05 significant. These thresholds were lower than that employed for identifying TSGs as the differences in gene expression within tissues between species are smaller than those between tissues within species [23]. In the differential expression analysis, the upregulated genes refer to genes that were upregulated in buffalo, whereas the downregulated genes were genes that were upregulated genes in cattle. We ranked genes according to their degree of differential expression (−log_10_p) from DEG analysis between buffalo and cattle. Subsequently, we selected the highest and lowest 10% of all orthologous genes as the most divergent and most conserved genes, respectively.

## 3. Results

### 3.1. Summary of Gene Expression Profiles in Buffalo

We analyzed 73 newly generated and 282 existing RNA-Seq samples, representing 57 tissues in domestic buffalo. Using a uniform pipeline of analysis, we generated ~9.27 billion clean reads. Details of sample information were summarized in Appendix A. We further divided these tissues into 20 classes following Yao et al. [23]. (Figure 1a). As expected, we observed a clear clustering of these tissues based on their gene expression patterns (Figure 1b and Appendix A). Some tissues, such as the brain, testes, and liver, formed a distinct cluster separating from other tissues (Figure 1c). Tissues with similar physiological functions exhibited greater correlation in their gene expression patterns, such as the small intestine and large intestine (Figure 1c).

### 3.2. Sequence Similarity of Genome and Transcriptome between Buffalo and Cattle

Based on the reference genomes of buffalo (UOA_WB_1) and cattle (ARS-UCD1.2), we analyzed the genomic, as well as transcriptomic sequence consistency between these two species. Our findings indicated that the buffalo reference genome had 98.87% of its sequences aligning with the cattle reference genome, with an average consistency of 95.88%. This agrees with a previous study [16]. Moreover, we compared the sequence similarity of CDS regions between the two species, revealing that 87.91% of buffalo CDS regions could be mapped to cattle CDS regions, with an average consistency of 98.12%. These findings demonstrated the relatively strong collinearity between the reference genome sequences of the two species, laying a foundation for subsequent comparative transcriptome analyses.

### 3.3. Conservation of Global Gene Expression Patterns between Buffalo and Cattle

Following a previous study [21], we classified genes into four categories: one-to-one homology, complex homology (including one-to-many, many-to-one, or many-to-many), no homology, and non-protein. In each tissue, we compared the proportion of expressed genes in each category to the total number of expressed genes (TMP > 0.1) (Appendix A). In Buffalo, the average proportions of one-to-one homology, complex homology, no homology, and non-protein genes across tissues were 82.62%, 7.94%, 4.07%, and 5.37%, respectively, while in cattle, they were 85.37%, 10.51%, 1.79%, and 2.33%. Similarly, we compared the summed gene expression (log_2_(TMP)) in these categories to the total expression (Appendix A). In buffalo, the average proportions of one-to-one homology, complex homology, no homology, and non-protein genes across tissues were 83.85%, 10.14%, 3.15%, and 2.84%, respectively, while in cattle, they were 83.85%, 12.54%, 1.59%, and 2.01%. Our results revealed that the genes in the four categories exhibited similar patterns in terms of both the number of expressed genes and their expression levels between the two species. Notably, the one-to-one homology genes had the largest number of expressed genes and represented the predominant expression levels in the tissues (Appendix A). Therefore, our subsequent analyses were based on 16,497 one-to-one orthologous genes for comparing the transcriptomes between the two species. 

To assess the conservation of gene expression between buffalo and cattle, we compared the number of expressed genes in each tissue and observed a significant correlation (Spearman’s r = 0.59, *p* = 0.0071) between the two species (Figure 2a). Notably, the testes exhibited the highest number of expressed genes in both species (n_buffalo_ = 15,042; n_cattle_ = 13,764), while the muscle (n_buffalo_ = 12,285; n_cattle_ = 11,182) and blood/immune tissues (n_buffalo_ = 12,230; n_cattle_ = 10,994) showed the lowest numbers of expressed genes.

To visualize the variation in gene expression among samples, we used the t-SNE-based method and found that samples from similar tissues clustered together rather than by species, indicating the conservation of gene expression among the species (Figure 2b,c). This observation was further supported by the hierarchical clustering of tissues based on the mean or median gene expression in each tissue (Appendix A). Additionally, we found that correlations based on gene expression in the same tissue between species were significantly higher than those observed between different tissues in the same species (Appendix A). Tissues, such as the liver, brain, small intestine, and stomach, exhibited the highest similarity in gene expression between buffalo and cattle, while tissues, such as skin and salivary gland, showed the lowest similarity (Appendix A). Finally, we observed that buffalo and cattle shared most genes at the top (highest expression) and bottom (lowest expression) 10% of genes sorted by their median level of expression in each tissue (Figure 2d).

### 3.4. Detection and Comparison of Tissue-Specific Genes (TSGs) 

By counting the number of tissues where a gene is expressed (TPM > 0.1), we found that genes tend to express ubiquitously (in all tissues) or tissue-specifically (in a few tissues) in both buffalo and cattle (Figure 3a). There was a significant correlation between the number of tissues where each gene was detected as expressed (TPM > 0.1) in each species (Spearman’s r = 0.87, *p* < 2.2 × 10^−16^), indicating global conservation of tissue-specific expression among orthologous genes. We then identified TSGs using the R package Limma described in a previous study [23], and genes with adjusted *p* value < 0.05 and log_2_(FC) > 1.5 were considered as TSGs. The number of TSGs in each tissue was significantly correlated between the two species (Spearman’s r = 0.48, *p* = 0.033) (Figure 3b). The testes exhibited the largest number of TSGs in both buffalo and cattle, while the salivary gland and the mammary gland displayed the smallest number in buffalo and cattle, respectively. Moreover, we discovered a significant overlap of TSGs in the same tissues between the two species (Hypergeometric test, FDR < 0.0001) (Figure 3c). Notably, the top 10 TSGs with the highest expression detected in buffalo exhibited a strong tissue specificity in cattle and vice versa for the top 10 TSGs in cattle (Figure 3d). These findings suggested that TSGs were conserved between buffalo and cattle.

In testes, TSGs uniquely expressed in buffalo were enriched in functions related to the regulation of response to DNA damage stimulus, regulation of DNA repair, RNA localization, ncRNA processing, and chromatin remodeling. In contrast, TSGs uniquely expressed in cattle were enriched in functions related to cell junction assembly, positive regulation of cell projection organization, axonogenesis, and synapse assembly (Figure 3e, Appendix A). Interestingly, the TSGs overlapping between the two species were enriched in functions related to cellular processes involved in reproduction in multicellular organisms, microtubule-based movement, cilium organization, germ cell development, and cilium assembly (Figure 3e, Appendix A).

### 3.5. Detection and Comparison of House-Keeping Genes (HKGs)

A method described by Zhang et al. [7] was used to explore HKGs in buffalo and cattle. We identified 8385 and 7923 preliminary HKGs (pHKGs) in buffalo and cattle, respectively, of which the median TPM was >1 in all tissues. Of these preliminary HKGs, 7491 (89.3% in buffalo and 94.5% in cattle) were found to be shared in both species. We further classified these shared pHKGs into three groups (high, medium and low) based on their expression variability across tissues, measured by the coefficient of variation (CV). We found that 1611 (21.5%), 3844 (51.3%) and 2036 (27.2%) pHKGs showed high, medium and low expression variability in buffalo, while 1702 (22.7%), 3821 (51.0%) and 1968 (26.3%) showed high, medium and low expression variability in cattle, respectively (Figure 4a). A total of 1211, 2567, and 1134 pHKGs showed consistently low, medium and high expression variability between buffalo and cattle, respectively (Figure 4a). GO enrichment analysis showed that highly variable genes were related to energy metabolism (e.g., sulfur compound metabolic process, small molecule catabolic process, fatty acid catabolic process and lipid catabolic process etc.), medium variable genes were related to basic biological activities (e.g., macroautophagy, DNA damage repair, peptidyl-lysine modification and stem cell population maintenance) and low variable genes were involved in organelle functions (e.g., mitochondrial translation, ribosome biogenesis, Golgi vesicle transport and protein insertion into membrane) (Figure 4b, Appendix A). 

Additionally, we considered 1211 genes that expressed with low variability across tissues as conserved HKGs and further divided them into three groups (low: TMP ≤ 10, medium: 10 < TPM ≤ 50, and high: TMP > 50) depending on their expression level. Our results indicated 98 (8.1%), 813 (67.1%) and 300 (24.8%) HKGs showed low, medium and high expression levels in buffalo, while in cattle, values were 122 (10.1%), 813 (67.1%) and 276 (22.8%), respectively (Figure 4a). Among these, 54 (4.5%), 673 (55.6%) and 204 (16.8%) HKGs demonstrated consistent low, moderate, and high expression levels across the two species, respectively (Figure 4a). Notably, the expression patterns of these genes were found to be similar between the two species (Figure 4c and Appendix A). Highly expressed HKGs were associated with Golgi vesicle transport, regulation of RNA splicing, cytoplasmic translational initiation and protein targeting. Conversely, medium and low-expression HKGs were related to the rRNA metabolic process, ribosome biogenesis, RNA catabolic process and RNA modification (Figure 4d, Appendix A).

### 3.6. Detection of Differentially Expressed Genes (DEGs) between Species

We identified DEGs between buffalo and cattle in each matching tissue. The brain showed the lowest number of DEGs, while the skin had the highest one (Figure 5a). We selected the top and bottom 10% of genes with the smallest and largest *p*-values from the differential expression analysis as the divergent and conserved genes between species, respectively, and compared these genes with TSGs. We found that TSGs in some tissues tended to be differentially expressed between species, such as skin, while those in other tissues tended to be conserved between species, such as mammary glands (Figure 5b). We conducted GO enrichment analysis for the DEGs in the skin between buffalo and cattle and found that those upregulated in buffalo were associated with epidermis development, skin development, and keratinocyte differentiation, while those upregulated in cattle were associated with collagen fibril organization, cell-substrate adhesion, and collagen metabolic processes (Figure 5c, Appendix A). These results presumably reflected differences in the morphology of skin between the two species.

## 4. Discussion

In this study, we integrated transcriptomic data from buffalo, established a comprehensive gene expression atlas, and performed a comparative transcriptomic analysis between buffalo and cattle. Samples in our transcriptomic dataset clustered by tissue in the expression heatmap, despite being generated from various breeds of river and swamp buffalo. This suggested that our integrated data were devoid of any conspicuous batch effects and, furthermore, underscored that expression differences between tissues exceeded those between breeds [29]. We identified TSGs of 20 tissues and found that these TSGs were mainly related to the physiological function of tissues, which also demonstrated the reliability of our results. This resource will enhance our understanding of the genetic and biological processes of complex traits in future studies [30].

Buffalo and cattle exhibited conservation in their overall expression patterns, TSGs, and HKGs. Firstly, the correlation of the numbers of genes expressed in each tissue between the two species is 0.59, which was consistent with the findings of a previous comparative transcriptome study in cattle and humans [23]. The correlation in expression patterns within the same tissue between the two species was much higher than that within the same species in different tissues, as previously reported [23]. This finding further confirmed the conservation of tissue expression patterns between species [29]. We identified a significant overlap of TSGs between species, indicating their conservation. Notably, the testes had the highest number of TSGs in both species and the shared TSGs between species, reflecting a relatively unique expression pattern [31,32,33]. We identified 8385 and 7923 pHKGs in buffalo and cattle, respectively, which was similar in mice [34]. Among these pHKGs, 89.3% in buffalo and 94.5% in cattle were shared between species. Moreover, 65.6% of shared pHKGs showed the same expression variation level, and 76.9% of shared HKGs exhibited the same expression level. This suggested that HKGs were conserved across species in terms of quantity, expression variation, and expression levels [7,35].

Despite the strong conservation of the transcriptome between buffalo and cattle, several genes that were differentially expressed between the two species have been identified. The largest number of differentially expressed genes was observed in the skin. GO enrichment analysis revealed that the upregulated genes in buffalo were associated with epidermis development, skin development, and keratinocyte differentiation, while those upregulated in cattle were associated with collagen fibril organization, cell-substrate adhesion, and collagen metabolic processes. Collagen fibrils are a critical component of animal skin [36,37,38], and the differential expression of these genes may be related to structural and functional differences between buffalo and cattle skin. For example, buffalo skin had a lower density of sweat glands and thicker skin than cattle [39], which could contribute to the observed differences in gene expression in skin tissue between the two species.

## 5. Conclusions

Our study provided a multi-tissue gene expression atlas and identified tissue-specific and housekeeping genes in buffalo. This enriches the functional annotation of the buffalo genome and establishes a foundation for further exploration of its genomic information and biological mechanisms underlying complex traits and adaptive evolution in buffalo. Additionally, we compared the conservation of transcriptomes between buffalo and cattle, which deepens our understanding of gene expression conservation between species and provides future direction for more comprehensive comparative analyses between the two species at a functional level.

## Figures and Tables

**Figure 1 genes-14-00890-f001:**
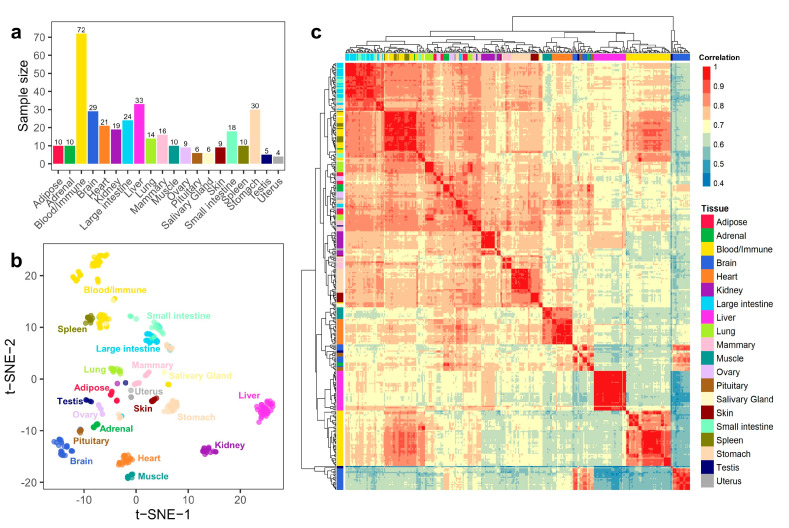
Gene expression profile among 20 tissue classes in buffalo. (**a**) Sample size per tissue class in buffalo. (**b**) The plot of t-SNE of samples based on gene expression. (**c**) Hierarchical clustering heat map of samples based on Pearson’s correlation coefficient for all genes. Color intensity indicates the correlation between tissues, red indicates a high correlation (1), and blue indicates a low correlation (0.4).

**Figure 2 genes-14-00890-f002:**
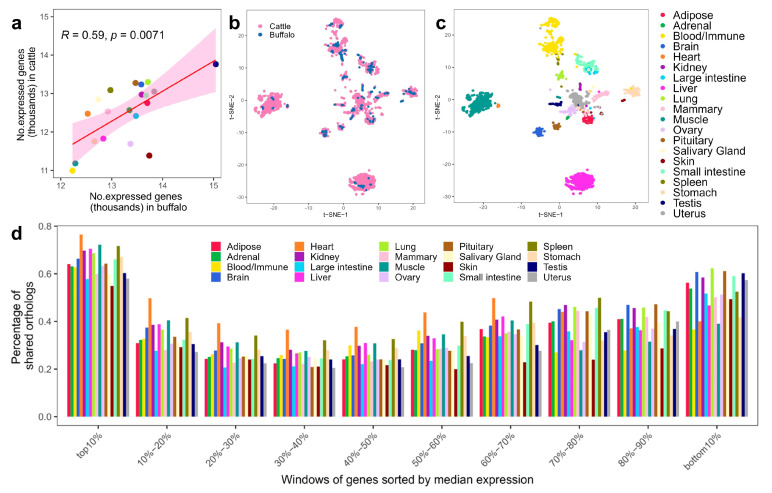
Conservation of transcriptomes of 20 common tissues in buffalo and cattle. (**a**) Spearman’s correlation of a number of expressed genes (median TPM > 0.1) across tissues between buffalo and cattle. Each dot represents a tissue. (**b**) The plot of t-SNE of samples based on batch-corrected gene expression (Methods). Each dot represents a sample colored by species types. (**c**) Same as in (**b**) but colored by tissue types. (**d**) Percentage of orthologous genes shared in each window between buffalo and cattle. Genes were ranked (from largest to smallest) by median expression in each tissue of each species and then divided into ten windows evenly.

**Figure 3 genes-14-00890-f003:**
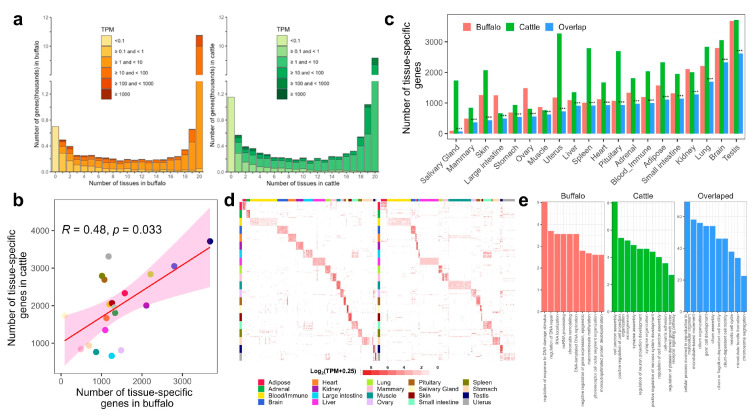
Comparison of tissue specificity of gene expression. (**a**) Gene expression levels and the number of tissues in which genes were expressed (median TPM > 0.1) in buffalo (**left**) and cattle (**right**). (**b**) Spearman’s correlation of a number of tissue-specific genes across tissues between buffalo and cattle. Each dot represents a tissue. (**c**) A number of tissue-specific genes (log_2_(fold-change) > 1.5 and FDR < 0.05) and their overlap across 20 tissues in buffalo and cattle (*** *p* < 0.001). (**d**) Expression profiles of top 10 tissue-specific genes that are detected in buffalo among both buffalo (**left**) and cattle samples (**right**). Each row represents a gene, and each column represents a sample from the corresponding tissue. The color represents log_2_-transformed expression value (log_2_(TPM + 0.25)). (**e**) Top 10 significantly enriched GO terms for tissue-specific genes (uniquely in buffalo or cattle and overlapped between buffalo and cattle) in testis.

**Figure 4 genes-14-00890-f004:**
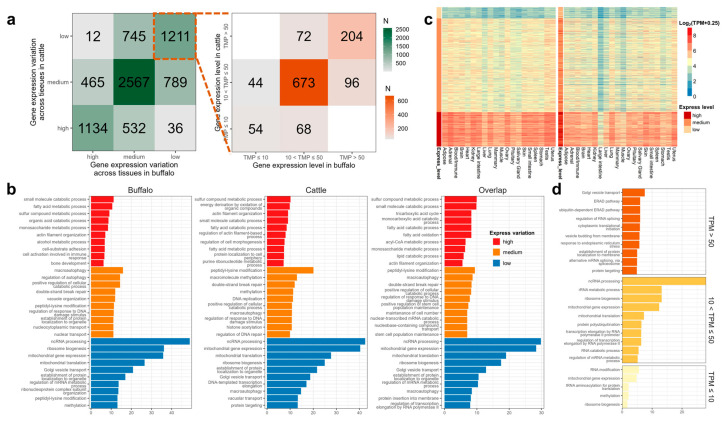
Comparison of housekeeping genes. (**a**) The number of low, moderate, and high expression variability HKGs (**left**), and the number of the low (TMP ≤ 10), medium (10 < TPM ≤ 50), and high (TMP > 50) expression levels of HKGs with low expression variability. (**b**) The GO enrichment of preliminarily screened HKGs. (**c**) Heatmap of gene expression of low-variable HKGs in buffalo (**left**) and cattle (**right**). (**d**) GO enrichment of overlapped low-variable HKGs.

**Figure 5 genes-14-00890-f005:**
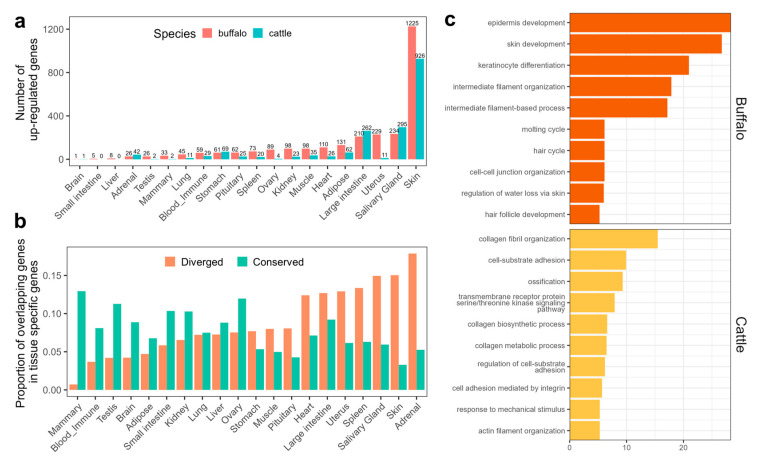
Comparison of average gene expression across 20 tissues between buffalo and cattle. (**a**) A number of significantly upregulated genes across tissues in buffalo (red) and cattle (green) using the cutoff of log_2_(FC) > 1.3 and FDR < 0.05. (**b**) The proportion of overlapping genes between tissue-specific genes and diverged genes (orange) and conserved genes (green) in the tissue-specific gene. (**c**) GO enrichment of significantly upregulated genes in Skin in buffalo and cattle.

## Data Availability

The multi-tissue gene expression atlas, tissue-specific genes and house-keeping genes of water buffalo generated in this study are publicly available at https://doi.org/10.6084/m9.figshare.22219327.v1, accessed on 6 March 2023. The raw RNA-Seq data for the 73 swamp buffalo samples have been deposited at the Sequence Read Archive (SRA) with study ID PRJNA951806. The accessions for the previously published datasets can be found in Appendix A.

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
