# Peer review of "A Multi-Tissue Gene Expression Atlas of Water Buffalo (*Bubalus bubalis*) Reveals Transcriptome Conservation between Buffalo and Cattle"

_genes, 2023, doi:10.3390/genes14040890_

Round 1

Reviewer 1 Report

The paper "A multi-tissue gene expression atlas of water buffalo (Bubalus bubalis) reveals transcriptome conservation between buffalo and cattle" by Si and colleagues describes the generation and analysis of 73 RNA-Seq samples from several tissues of the domestic Buffalo. The authors procede also with the addition of public RNA-Seq data of Buffalo and Cattle samples. The paper provides a new complete transcriptomics atlas for Bubalus bubalis, and also a summarized analysis of its features and comparison with a relative species. While potentially an interesting paper, it currently suffers from several issues in clarity and biological relevance, which call for, in my opinion, an extensive rewriting, restructuring and revision. Below, my points.

- The dataset is said to be provided as an attachment on Figshare, but it is currently under embargo, so I did not have a chance to review it. One issue is that it seems the dataset is provided as TPM data, and not as FASTQ data. A more important issue is that Figshare is not the appropriate location for this kind of study, because it is not searchable and does not apply the standards of quality of databases designed for this. The authors must upload their dataset to a public RNA-Seq repository, like Gene Expression Omnibus or ArrayExpress. The authors must recognize that all the public Buffalo data they used for comparison are deposited on NCBI, and if all those studies had used Figshare, the data wouldn't be available for this study.

- Line 97. It is completely unclear to me what the comparison for calculating differential expression is. The authors show a number of differentially expressed genes for each tissue and for both buffalo and cattle. But what is the reference tissue? The authors define the term "TSG" as "tissue-specific gene", calculated by limma. Limma is a famous package to perform differential expression, but again: what is the reference tissue? Is it used in a "one tissue vs. all other tissues" setup, each time a tissue is selected?

- Line 129: what is the technical meaning of "clean" reads? Does it mean reads that survived trimming? Or reads that were correctly aligned on the Buffalo genome? Please expand or use a more descriptive term.

- Line 133: what does it mean that some tissues show "independent expression"? Again, the terminology is really vague. From the referenced data dimensionality figure 1b, it doesn't seem that the mentioned tissues show any peculiar pattern. So I would remove the "independent" part, or clarify what it means.

- Figure 1b: the coloring is misleading, as there are too many (and too similar) colors for each of the sequenced tissues. The authors should use a different method to clarify the image, a few suggestions: using labels, using a combination of different shapes and symbols. Increasing the size of figure 1b, making it an independent TSNE plot, and also adding a separate legend would help. Also, not all 57 tissues are shown, and I believe the authors should indicate also the number and TSNE projection for the tissues with a lower amount of samples, to understand their location in the general histological space. Again, having a larger TSNE image would help in fitting all the 57 tissues (currently, only ~20 are shown).

- Line 85: what is the rationale of orthology inference performed by OrthoFinder? Is it based on a bidirectional best-hit principle? Currently, consensus methods are favored (like, DIOPT, or babelgene), however this could not be available for Bubalus bubalis. The authors nevertheless should explain the rationale of the chosen orthology conversion tool.

- In general, the authors oversimplify the task of comparing two different species by simple orthological conversion. For example, the authors should specify the percentage genome AND transcriptome similarity between Buffalo and Cattle. Also, they should describe how complex the sequence orthology between the two species is. For example, the authors keep 16497 genes with a simple one-to-one orthologous correspondence. This will leave out roughly 5000 genes with other types of orthology (one-to-many, many-to-one or many-to-many). To dismiss this complexity is also to miss one key element in the comparison between two species, and the authors should provide a thorough analysis of it.

- Line 87: the function IntegrateData was designed for single-cell data. Why do the authors use it for bulk data? Assuming this is correct, what are the hidden confounding factors that were removed? Is this a glorified batch correction? And if so, why was it applied? It does not seem a standard operation for the analysis of a single dataset, which should not contain hidden variables. If the authors found hidden variables, what are they?

- The resolution in many figures is so low that they are unreadable. I urge the authors to produce a manuscript with high resolution figures, or larger text (e.g. in Figure 4), otherwise this paper will be yet another unusable addition to the pile of blurry transcriptomics publications.

- The authors define HouseKeeping Genes and show them in figure 4c. While this is a nice summary heatmap, it does not show any of the genes, and so it is not useful for molecular biologists who wish to select, for example, tissue-specific HK genes or general genes. The authors should make another heatmap showing the readable gene symbols (so, 20 or 30 at most) of the most robust and most highly expressed HKGs. Another way to do this is to show average expression and variance of each gene in the Buffalo dataset, as a scatter plot, in order to show the most robust housekeeping genes as those with the highest expression and the lowest variance.

- On line 118, the authors specify that differential expression has been performed between species (Buffalo and cattle). But if so, how come the differential expression is shown for both Buffalo and cattle (Figure 5)? Are the authors reporting only up-regulated genes, in order to show as "up-regulated in cattle" genes that result as downregulated in the comparison buffalo vs. cattle? If so, the authors must clarify this.

- Figure S3 reports the correlation Cattle-Buffalo for all orthologous genes in several tissues. While interesting, this plot only reports the Pearson correlation coefficient. The authors must perform this analysis in a more appropriate way, such as: (1) reporting the scatter plots for each tissue, (2) reporting both the Pearson and Spearman correlation coefficients, (3) reporting the p-value of the correlation. The risk with reporting only correlation coefficients is that they may hide issues like outliers, low sample size, or peculiar distributions. Ideally (but optionally) the authors should report also which genes differ the most in each tissue scatter plot between the two species.

Reviewer 2 Report

This is an interesting and well-written manuscript aimed to study a multi-tissue gene expression atlas for water buffalo and to compare it with cattle. All sections of the manuscript are very clear and well supported. I only suggest considering next several minor corrections to improve the quality of the manuscript:

-       Line 108: Remove the word “have”.

-       Line 112: Remove the word “have”.

-       Line 119: Why did you use a different log2(FC) that the one you used for TSGs in line 101?

-       Line 130: Remove “(2022)” and complete the sentence indicating, what was followed from Yao et al.?

-       Line 171: Replace “share” by “shared”.

-       Line 212: Remove “(2022)”.

-       Line 220: Add the word “respectively” at the end of the sentence.

-       Line 226: Replace “modification, stem” by “modification and stem”.

-       Line 228: Replace “transport, protein” by “transport and protein”.

-       Line 239: Replace “show” by “showed”.

-       Line 241: Add “respectively” at the end of the sentence.

-       Line 242: Add “respectively” at the end of the sentence.

-       Line 261: Replace “reflect” by “reflected”.

-       Line 273: Replace “suggest” by “suggested”.

-       Line 275: Replace “underscores” by “underscored”.

-       Line 275: Replace “exceed” by “exceeded”.

-       Line 277: Replace “demonstrate” by “demonstrated”.

-       Line 280: Replace “exhibit” by “exhibited”.

-       Line 282: Replace “is” by “was”.

-       Line 284: Replace “is” by “was”.

-       Line 285: Replace “confirms” by “confirmed”.

-       Line 290: Replace “is” by “was”.

-       Line 293: Replace “suggest” by “suggested”.

-       Line 294: Replace “are” by “were”.

-       Line 297: Replace “are” by “were”.

-       Line 299: Replace “are” by “were”.

-       Line 301: Replace “are” by “were”.

-       Line 303: Separate square bracket from the text.

-       Line 305: Replace “has” by “had”.

-       Line 341: Remove the period sign at the beginning of the reference.

-       Line 342: Journal title should be italic, Journal date should be bold, and Journal volume should be italic style. Please make the same corrections to all references.

Round 2

Reviewer 1 Report

I think the authors answered all my comments, and I especially thank them for sharing their FASTQ data on the NCBI database. I am sure this will help future scientists working on mammalian transcriptomics.